# The SGLT2 inhibitor empagliflozin improves cardiac energy status via mitochondrial ATP production in diabetic mice

Jungmi Choi[1,7], Naoki Matoba[1,7], Daiki Setoyama[2], Daiki Watanabe[1], Yuichiro Ohnishi[1], Ryuto Yasui[1], Yuichirou Kitai[3], Aki Oomachi[1], Yutaro Kotobuki[4], Yoichi Nishiya[4], Michael Paul Pieper [5], Hiromi Imamura[6], Motoko Yanagita [3] & Masamichi Yamamoto [1,3✉]

Empagliflozin, a sodium-glucose co-transporter 2 inhibitor developed, has been shown to reduce cardiovascular events in patients with type 2 diabetes and established cardiovascular disease. Several studies have suggested that empagliflozin improves the cardiac energy state which is a partial cause of its potency. However, the detailed mechanism remains unclear. To address this issue, we used a mouse model that enabled direct measurement of cytosolic and mitochondrial ATP levels. Empagliflozin treatment significantly increased cytosolic and mitochondrial ATP levels in the hearts of *db/db* mice. Empagliflozin also enhanced cardiac robustness by maintaining intracellular ATP levels and the recovery capacity in the infarcted area during ischemic-reperfusion. Our findings suggest that empagliflozin enters cardiac mitochondria and directly causes these effects by increasing mitochondrial ATP via inhibition of NHE1 and Nav1.5 or their common downstream sites. These cardioprotective effects may be involved in the beneficial effects on heart failure seen in clinical trials.

[1] Department of Research Promotion and Management, National Cerebral and Cardiovascular Center, Kishibe-Shimmachi, Suita, Osaka 564-8565, Japan. [2] Department of Clinical Chemistry and Laboratory Medicine, Kyushu University Hospital, Fukuoka, Japan. [3] Department of Nephrology, Kyoto University Graduate School of Medicine, Kyoto University, Shogoin-Kawahara-cho, Sakyo-ku, Kyoto 606-8507, Japan. [4] Medicine Division, Nippon Boehringer Ingelheim Co., Ltd., 2-1-1 Osaki, Shinagawa-ku, Tokyo 141-6017, Japan. [5] CardioMetabolic Diseases Research, Boehringer Ingelheim Pharma GmbH & Co. KG, Birkendorfer Strasse 65, Biberach an der Riss 88397, Germany. [6] Department of Functional Biology, Graduate School of Biostudies, Kyoto University, Yoshida-konoe-cho, Sakyo-ku, Kyoto 606-8501, Japan. [7] These authors contributed equally: Jungmi Choi, Naoki Matoba. ✉email: yamamoto.mailserver@gmail.com

Sodium-glucose cotransporter 2 (SGLT2) inhibitors are a unique class of oral antidiabetic medications that reduce glucose reabsorption in the renal proximal tubules, thereby enhancing urinary glucose excretion[1]. In the EMPA-REG OUTCOME trial, a large randomized controlled clinical trial, the highly selective SGLT2 inhibitor empagliflozin significantly reduced the risk of three-point major adverse CV events, cardiovascular death, heart failure hospitalization and composite renal outcomes in patients with type 2 diabetes (T2D) with established cardiovascular disease[2].

Multiple hypotheses have been proposed to explain the beneficial effects of SGLT2 inhibitors[3,4], which can be multifactorial and acute/chronic[5]; the most common one is via an effect on diuresis/natriuresis, but the mechanisms involved in these impressive cardiac benefits are incompletely understood. Several research groups have suggested the potential efficacy of empagliflozin on cardiac energetics. Changes in cardiac energy production play a critical role in the pathophysiology of heart failure. The failing heart faces an energy deficit primarily because of a decrease in mitochondrial oxidative capacity, which is partly compensated for by an increase in ATP production from glycolysis occurring in the cytoplasm[6]. Several studies have shown that empagliflozin improved cardiac energy production in animal experiments[7,8]. Indeed, a single dose of empagliflozin in fasting diabetic *db/db* mice was associated with an improvement in cardiac energetics, which is also associated with an increase in ketone levels[7]. Similarly, Verma et al. reported that chronic administration of empagliflozin enhanced ATP production in the heart of *db/db* mice, although they argued that this was due to an increase in glucose and fatty acid oxidation rather than the utilization of ketone bodies[8]. To investigate the mechanism underlying the improvement in cardiac energy status with empagliflozin, real-time and accurate monitoring of the pathways and amount of energy production in vivo is needed.

Recently, using a modified *Ateam* (GO-ATeam[9]), which can directly quantify intracellular ATP levels via Fluorescence Resonance Energy Transfer (FRET) in cytosol and mitochondria, we generated transgenic mice to monitor subcellular ATP levels in the whole body, organ and cells, as well as in the beating heart (cytoATP-Tg for cytosol; mitoATP-Tg for mitochondria; manuscript submitted). These mice enabled us to visualize ATP levels in real time and in a living state by measuring the FRET/GFP ratio and proved ideal for studying drugs that affect energy metabolism.

In the present study, we crossed cytoATP-Tg or mitoATP-Tg mice with a mouse model of T2D and assessed the effects of empagliflozin on the cardiac energy status of those mice. In addition, several studies have demonstrated the beneficial effects of empagliflozin on myocardial infarction (MI) in diabetic animal models[10,11], prompting us to investigate the real-time ATP change in cardiac energy production in an ischemic-reperfusion model of MI. Furthermore, to gain insight into the direct effects of empagliflozin on the myocardium and its underlying molecular pathway, we also conducted experiments using isolated cardiomyocytes.

## Results

**Chronic treatment with empagliflozin reverses the energy reduction due to diabetic heart disease in mice with type 2 diabetes mellitus.** Using ATeam mice, we first investigated whether or not it was possible to visualize the decrease in cardiac ATP levels when T2D is present and the attenuation of the decrease by empagliflozin. After eight-week administration of empagliflozin, blood glucose levels in *db/db* mice were significantly decreased compared to controls (Supplementary Fig. 1a, b). Blood concentrations of ketone bodies were increased in *db/db* mice compared to wild-type mice, but no significant difference was observed between the empagliflozin administration *db/db* group and the control *db/db* group (Supplementary Fig. 1c, d). In addition, body weight changes of empagliflozin-treated *db/db* mice were not significantly different from the weight change in the control *db/db* group (Supplementary Fig. 1e). The systolic blood pressure also did not differ markedly among the three groups (Supplementary Fig. 1f, g).

Next, we examined the ATP dynamics in the hearts of mice administered empagliflozin for 10 weeks by measuring the FRET/GFP ratio using a fluorescence microscope (Fig. 1a). First, we investigated the effect of autofluorescence in the heart on the measurement of FRET (ET470/40 nm, D570/40 nm) and GFP (ET470/40 nm, D515/30 nm). The autofluorescence intensity of FRET and GFP in the hearts were $2.97 \pm 0.006$ and $5.45 \pm 0.01$, respectively (Supplementary Fig. 2i–m). In cytoATP-Tg mice, the fluorescence intensity of FRET and GFP in the hearts were

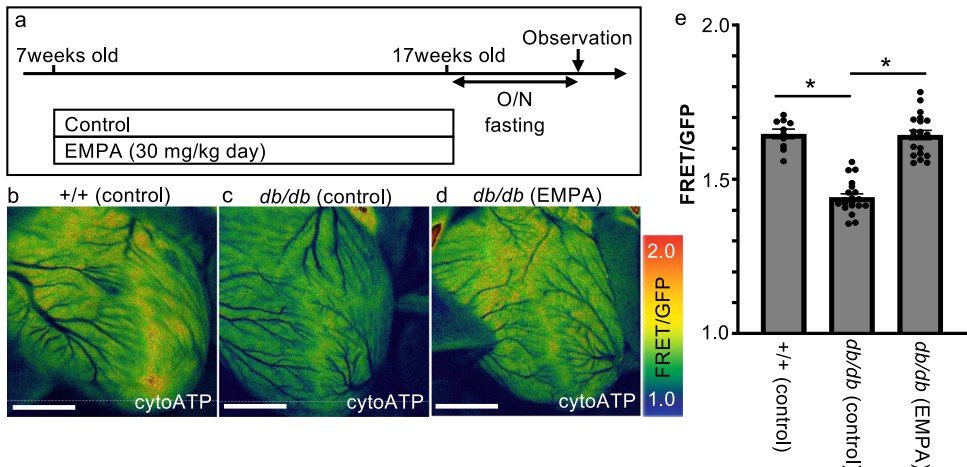

**Fig. 1 Chronic treatment with empagliflozin reverses cytosolic ATP loss in the heart of T2D model mice. a** Schematic illustration of the treatment regimen (EMPA empagliflozin). **b–d** Representative fluorescence images of a live heart from each condition. Warmer colors indicate higher ATP concentrations. Scale bar: 2 mm. **e** FRET/GFP ratios indicating the amount of ATP in the heart under each condition ($+/+$ [control]: $n = 10$, *db/db* [control]: $n = 20$, *db/db* [EMPA]: $n = 20$). $*p = 0.000$.

267.87 ± 0.71 and 162.16 ± 0.26, respectively (Supplementary Fig. 2a–d, m). These results indicate that the effect of autofluorescence in the heart was small enough to be negligible (Supplementary Fig. 2m).

The FRET/GFP ratio (ATP level) in the heart was significantly reduced to 1.44 ± 0.01 (about 1.03 mM) in *db/db* mice compared to 1.65 ± 0.01 (about 1.21 mM) in wild-type mice (Fig. 1b, c, e). This ATP reduction in *db/db* mice was not observed after 10 weeks of empagliflozin administration (1.64 ± 0.01, Fig. 1d, e).

**Chronic treatment with empagliflozin is cardioprotective by maintaining ATP levels in the infarcted region in type 2 diabetic hearts.** The increase in ATP levels induced by empagliflozin implies an improvement in energy maintenance in the heart. We therefore hypothesized that the beneficial effect of empagliflozin on an ischemia/reperfusion model[10], which is a severely stressed condition, might be mediated by the maintenance of the ATP levels in the heart. Thus, we investigated the effect of empagliflozin on the temporal changes in ATP levels in the infarcted myocardium, particularly during ischemic-reperfusion.

*db/db*; cytoATP-Tg mice were treated with control or empagliflozin for 10 weeks, starting at 7 weeks old, and at 17 weeks old, they underwent the ischemic/reperfusion procedure, and the ATP levels at the infarct sites of the heart were measured over time (Fig. 2a). Specifically, ischemia was induced by ligating the left anterior descending artery while monitoring the ECG after opening the chest with respiration maintained with a ventilator. Reperfusion was then initiated by removing the thread that had ligated the left anterior descending artery 30 min after the start of ischemia. In the *db/db* control group, a reduction in the ΔFRET/GFP ratio of about −0.28 (indicating a decreased ATP level) was induced in the infarct region compared to before ligation after 20 min of ligation (Fig. 2b, c, h). In the empagliflozin group, the ΔFRET/GFP ratio in the myocardial infarct region decreased by about −0.06 (Fig. 2e, f, h). This indicates that empagliflozin-treated mice had a higher ATP level in the infarcted area than the *db/db* control group. In the *db/db* control group, the ΔFRET/GFP ratio of the myocardial infarct scar gradually recovered up to 30 min after recanalization, and the ΔFRET/GFP ratio recovered to about −0.21 (Fig. 2d, h). In the empagliflozin-treated group, ATP levels were restored to baseline levels in the infarct region. (Fig. 2g, h). These results indicate that not only was the decrease in the ATP levels in the myocardial infarct region suppressed in the empagliflozin group compared to the *db/db* control group, but empagliflozin also maintained and restored energy, with a quick ATP recovery to the baseline levels after recirculation compared to the *db/db* control group (Fig. 2h). This difference suggests that empagliflozin confers "robustness" in ATP maintenance/production in the diabetic heart, and that this may contribute to the protective effects of empagliflozin against the ischemic-reperfusion injury as reported previously.

**Chronic treatment with empagliflozin reverses the decrease in mitochondrial ATP levels in the hearts of T2D model mice.** Since chronic administration of empagliflozin suppressed the decrease in cytosolic ATP levels in diabetic heart, we decided to measure the ATP levels in mitochondria, the major source of ATP production. In mitoATP-Tg mice, the fluorescence intensity of FRET and GFP in the heart was 392.6 ± 0.84 and 267.4 ± 0.30, respectively (Supplementary Fig. 2e–h, m). These results indicate that in mitoATP-Tg mice, the effect of autofluorescence in the heart was small enough to be negligible (Supplementary Fig. 2m).

*db/db*; mitoATP-Tg mice were treated with control or empagliflozin for 10 weeks, starting at 7 weeks old, and the

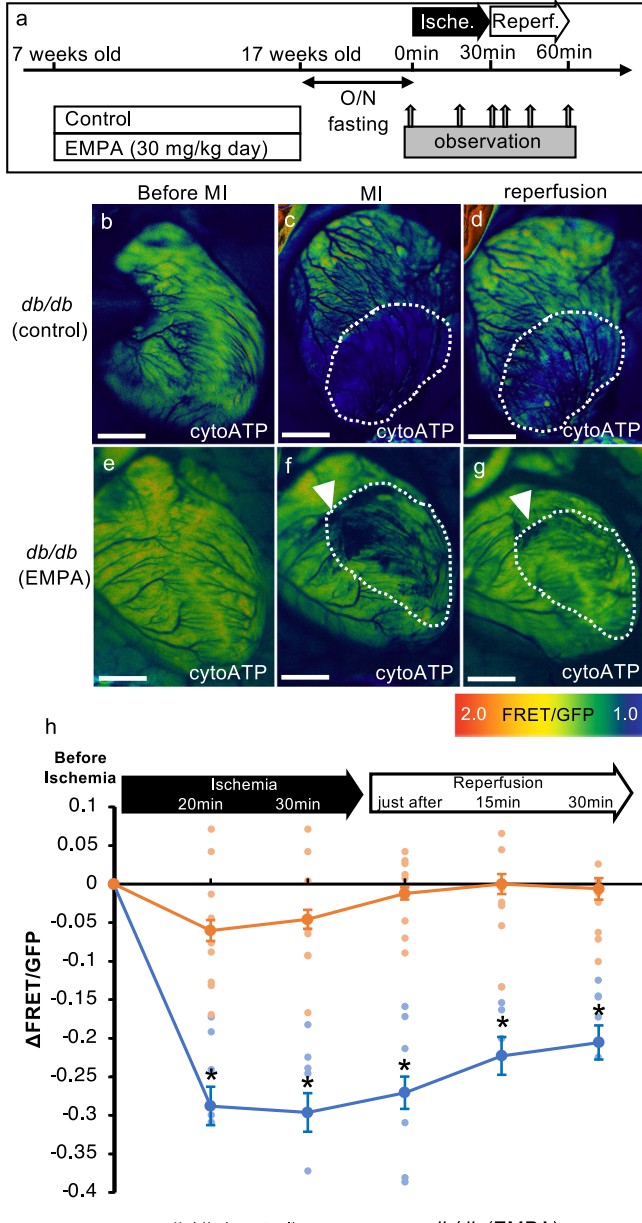

**Fig. 2 Chronic treatment with empagliflozin is cardioprotective by maintaining cytosolic ATP levels in the infarcted region of T2D hearts. a** Schematic illustration of the treatment regimen (EMPA empagliflozin). **b**–**g** Fluorescence images of a live heart from each condition (EMPA empagliflozin). Arrowheads show the sites of punctuation with a suture needle to induce ischemia. Dotted lines show the infarct regions. Warmer colors indicate higher ATP concentrations. Scale bar: 2 mm. **h** Time course changes in FRET/GFP ratios in the ischemic region during ischemia and reperfusion (*db/db* [control]: n = 6, *db/db* [EMPA]: n = 9; ischemia 20 min: *p = 0.001, ischemia 30 min: *p = 0.000, just after reperfusion: *p = 0.000, 15 min after reperfusion: *p = 0.001, 30 min after reperfusion: *p = 0.001).

amount of ATP in the heart mitochondria was measured at 17 weeks old (Fig. 3a). The FRET ratio was around 1.74 ± 0.01 in the wild-type mice but was significantly reduced to around 1.42 ± 0.01 in *db/db* with control treatment (Fig. 3b, c).

We then measured the amount of ATP in the cardiac mitochondria in a mouse model of T2D treated with empagliflozin. The mitochondrial ATP in the heart increased compared to the

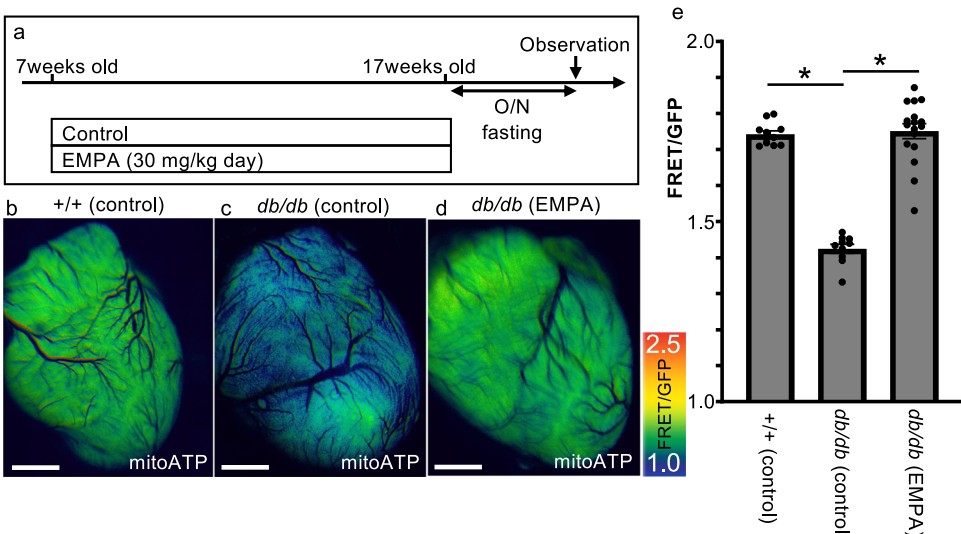

**Fig. 3 Chronic treatment with empagliflozin reverses mitochondrial ATP loss in the heart of T2D model mice. a** Schematic illustration of the treatment regimen (EMPA empagliflozin). **b–d** Representative fluorescence images of a live heart from each condition (EMPA empagliflozin). Warmer colors indicate higher ATP concentrations. Scale bar: 2 mm. **e** FRET/GFP ratios indicating the mitochondrial ATP level in the heart under each condition (+/+ (control): $n = 11$, $db/db$ (control): $n = 10$, $db/db$ (EMPA): $n = 17$). $*p = 0.000$.

$db/db$ control group, measuring about $1.75 \pm 0.01$ (FRET/GFT ratio), which is almost the same as that observed in the wild-type mice (Fig. 3d, e). These results indicate that empagliflozin suppressed the decrease in the mitochondrial ATP level in diabetic hearts.

**Single treatment of empagliflozin promotes an increase in mitochondrial ATP levels.** Next, we examined the timing of empagliflozin-induced changes in mitochondrial energy production.

In 8-week-old $db/db$; mitoATP-Tg mice, we found that the ATP level in the mitochondria was already lower than in wild-type mice (Fig. 4b). Therefore, we orally administered 30 mg/kg body weight of empagliflozin after 4 h of fasting and measured the ATP levels in the cardiac mitochondria of the mice in vivo 3 h later (Fig. 4a). Glucose and ketone concentrations in the blood were measured before measuring the ATP levels in the heart. Glucose levels were significantly lower in the empagliflozin group (approximately 135 mg/dL) than in the $db/db$ control group (approximately 422 mg/dL) (Supplementary Fig. 3a). In contrast, the ketone concentration was significantly higher in the empagliflozin group (approximately 6.9 mmol/L) than in the $db/db$ control group (approximately 2.35 mmol/L) (Supplementary Fig. 3b). At the time of observation, the mice treated with empagliflozin showed elevated mitochondrial ATP levels in the heart (Fig. 4b–d). At this time, empagliflozin was detected and quantified in mitochondria in the heart and other organs, except for the brain (Fig. 4h). These results suggest that empagliflozin entered cardiac mitochondria and increased the ATP levels in vivo as early as 3 h after administration. However, since the blood ketone concentration, which is considered an energy source for the heart, also increased during this period (Supplementary Fig. 3b), we were unable to determine whether the increase in the cardiac mitochondrial ATP level was a direct effect of empagliflozin or a secondary effect due to an increase in ketone concentrations.

Therefore, we isolated mature cardiomyocytes from 8-week-old $db/db$; mitoATP-Tg mice and examined the mitochondrial ATP levels in the same cells before and 1 h after the addition of empagliflozin. The results showed that the FRET/GFP ratio was $1.05 \pm 0.019$ in the control group, $1.25 \pm 0.020$ in the group with

10 nM empagliflozin, $1.22 \pm 0.020$ in the group with 100 nM empagliflozin, $1.31 \pm 0.015$ in the group with 500 nM empagliflozin and $1.40 \pm 0.013$ in the group with 1000 nM empagliflozin (Fig. 4e–g, $p = 0.0010, 0.0058, 0.0000, 0.0000$ vs. $db/db$ control). At this time, the ATP level in the cytoplasm was not changed (Supplementary Fig. 4a–e). This result indicates that empagliflozin directly increased the ATP level in mitochondria in cardiomyocytes apart from an increase in blood ketone concentrations.

**Empagliflozin maintains mitochondrial ATP levels in mature cardiomyocytes even under hypoxic conditions, demonstrating its cardioprotective effect.** Next, we examined the effect of empagliflozin on the temporal changes in mitochondrial ATP levels while trying to mimic MI.

Therefore, we isolated mature cardiomyocytes from 18-week-old $db/db$; mitoATP-Tg mice and examined the mitochondrial ATP levels in the cells over time during hypoxia exposure and oxygen recovery (Fig. 5a). The results showed that a significant reduction in the ΔFRET/GFP ratio of about $-0.11 \pm 0.01$ (indicating a decreased ATP level) was induced in mitochondria after 30 min of hypoxia compared to before the exposure (Fig. 5b, c, h). At 30 min after the induction of hypoxia, the cardiomyocytes were allowed to recover to high-oxygen conditions. However, the ΔFRET/GFP ratio had scarcely recovered even 30 min after exposure to an oxygen-rich environment, reaching about $-0.09 \pm 0.01$ (Fig. 5d, h). We also conducted the same experiments with mature cardiomyocytes treated with empagliflozin for 1 h. In the empagliflozin group, the ΔFRET/GFP ratio in mitochondria decreased by about $0.03 \pm 0.02$ after 30 min of exposure to low-oxygen conditions (Fig. 5e, f, h). Furthermore, mitochondrial ATP levels were restored to the baseline levels ($0.02 \pm 0.01$) by 30 min after exposure to high-oxygen conditions (Fig. 5g, h). These results indicate that not only was the decrease in mitochondrial ATP levels during low-oxygen exposure suppressed in the empagliflozin group compared to the control group, but empagliflozin also maintained and restored energy, with an increased rate of mitochondrial ATP recovery after exposure to an oxygen-rich environment compared to the control group (Fig. 5h).

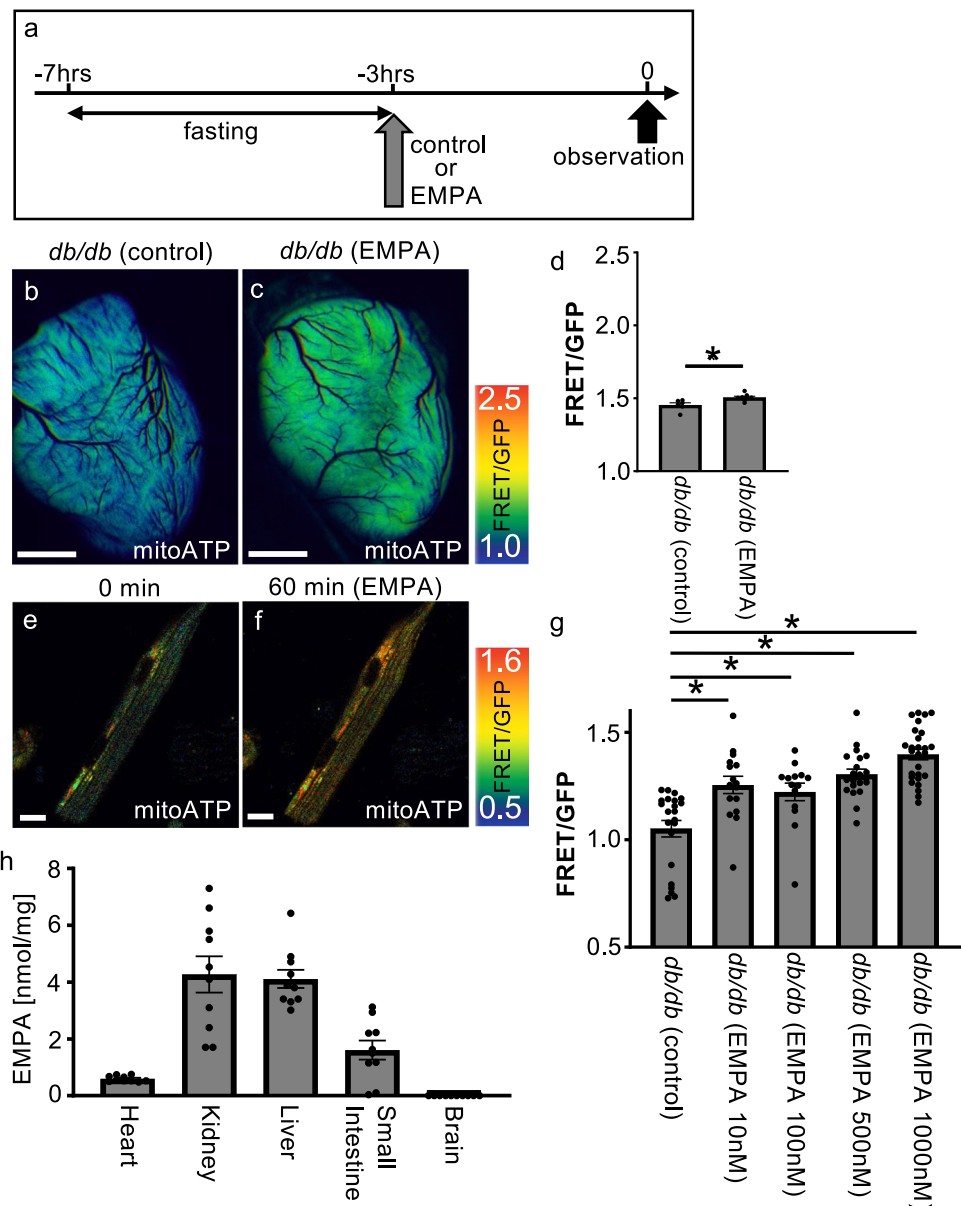

**Fig. 4 Single treatment with empagliflozin increases mitochondrial ATP levels in the heart of T2D model mice. a** Schematic illustration of the treatment regimen (EMPA empagliflozin). **b**, **c** Representative fluorescence images of a live heart from each condition (EMPA empagliflozin) with or without increased ATP. Warmer colors indicate higher ATP concentrations. Scale bar: 2 mm. **d** FRET/GFP ratio indicating the amount of ATP in the heart under each condition (*db/db* [control]: $n = 6$, *db/db* [EMPA]: $n = 8$). *$p = 0.013$. **e**, **f** Representative images of mitochondrial ATP concentrations before (**e**) and 1 h after (**f**) empagliflozin 1000 nM administration. Scale bar: 25 μm. **g** Graph of the FRET/GFP ratio in mature cardiomyocytes with control ($n = 22$), empagliflozin 10 nM ($n = 16$), empagliflozin 100 nM ($n = 14$), empagliflozin 500 nM ($n = 21$) and empagliflozin 1000 nM ($n = 27$) (*$p = 0.0010$, 0.0058, 0.0000, 0.0000, respectively). **h** Biodistribution of empagliflozin in mitochondria in various tissues at 3 h after empagliflozin administration ($n = 10$). Heart: 0.56 ± 0.029, kidney: 3.95 ± 0.581, liver: 3.85 ± 0.216, small intestine: 1.15 ± 0.315, brain: 0.0 ± 0.000 nmol/mg.

**Empagliflozin promotes increased mitochondrial membrane potential and mitochondrial ATP levels via Na+/H+ exchanger and Nav1.5.** It was shown that empagliflozin may directly promote an increase in mitochondrial ATP levels in cardiomyocytes, but the mechanism of action is unclear. Therefore, we investigated whether or not the increase in mitochondrial ATP levels was accompanied by an increase in mitochondrial membrane potential, which is of critical importance in maintaining the function of the respiratory chain to produce ATP.

The mitochondrial membrane potential was measured after adding vehicle or empagliflozin to mature cardiomyocytes isolated from 8-week-old wild-type mice, and 1 h later, JC-1

was applied to the cells (0.30 ± 0.004: $N = 120$, 0.33 ± 0.010: 135 each). The results showed that the mitochondrial membrane potential was significantly increased in the cardiomyocytes treated with empagliflozin (Fig. 6a–c, $p = 0.03$).

Empagliflozin reportedly inhibits not only SGLT2 but may also directly inhibit Na$^+$/H$^+$ exchanger 1 (NHE1)[12,13] and Nav1.5[14]. To investigate the possibility that these ion channels influence mitochondrial ATP levels in mature cardiomyocytes, we next performed experiments in which we added each inhibitor to mature cardiomyocytes isolated from 8-week-old +/+; mitoATP-Tg mice and incubated them with either vehicle or empagliflozin. The ATP levels in the mitochondria were measured 1 h after the

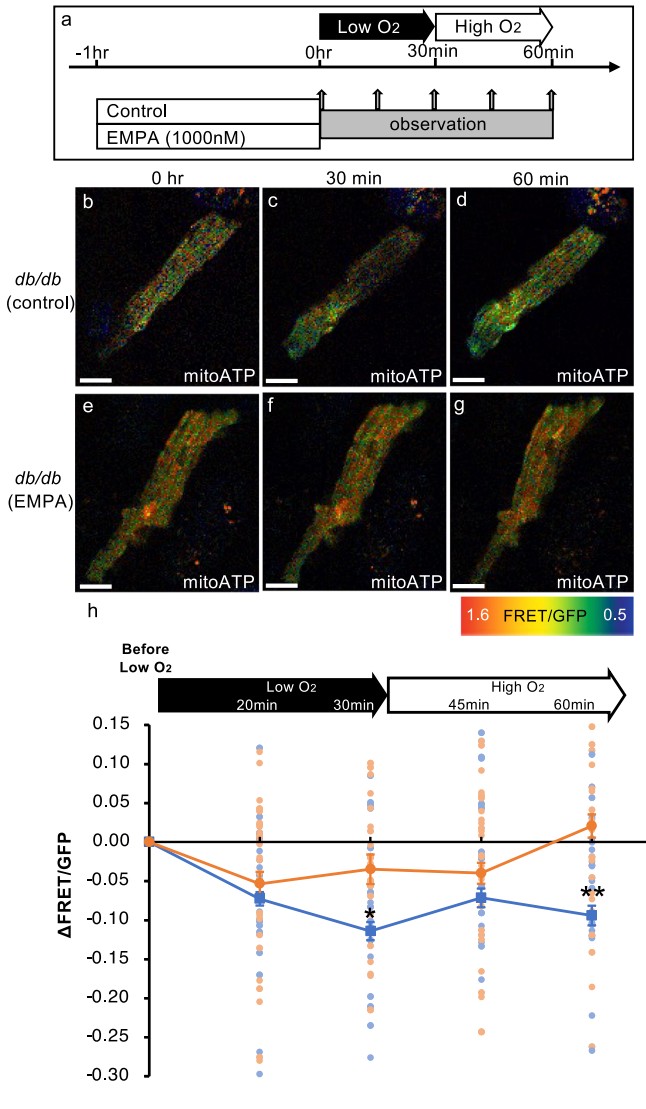

**Fig. 5 Single treatment of empagliflozin is cardioprotective by maintaining mitochondrial ATP levels in hypoxia in mature cardiomyocytes. a** Schematic illustration of the treatment regimen (EMPA empagliflozin). **b–g** Fluorescence images of cardiomyocytes from each condition (EMPA empagliflozin). Warmer colors indicate higher ATP concentrations. Scale bar: 25 μm. **h** Time course changes in FRET/GFP ratios in the cardiomyocytes under low- and high-oxygen conditions (*db/db* [control]: $n = 44$, *db/db* [EMPA]: $n = 40$; 30 min after exposure to low-oxygen conditions: $*p = 0.020$, 30 min after exposure to high-oxygen conditions: $**p = 0.000$).

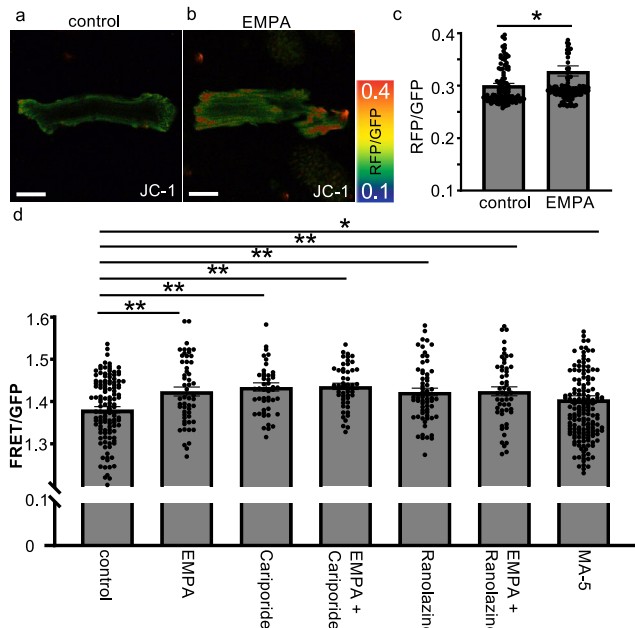

**Fig. 6 Empagliflozin promotes mitochondrial activity via the NHE1 and Nav1.5 pathways. a, b** Measurement of mitochondrial membrane potential using JC-1. Representative cellular images from each condition (EMPA empagliflozin). Warmer colors indicate higher mitochondrial membrane potential. Scale bar: 25 μm. **c** Graph of the mitochondrial membrane potential of mature cardiomyocytes with control ($n = 120$), empagliflozin 10 nM ($n = 134$). $*p = 0.021$. **d** Graph of the FRET/GFP ratio in mature cardiomyocytes isolated from 8-week-old mitoATP-Tg mice 1 h after addition of control ($n = 67$), 300 nM empagliflozin ($n = 54$), 10 μM cariporide ($n = 49$), cariporide and empagliflozin ($n = 49$), 30 μM ranolazine ($n = 65$), ranolazine and empagliflozin ($n = 53$) or 10 μM MA-5 ($n = 163$) ($**p = 0.0012, 0.0001, 0.0000, 0.0006, 0.0011, *p = 0.046$, respectively).

not the addition of empagliflozin to these inhibitors had an add-on effect. Consequently, the FRET/GFP ratio was $1.42 \pm 0.004$ when cariporide plus empagliflozin was added, showing no significant difference from cariporide alone ($1.42 \pm 0.006$) or empagliflozin alone ($1.42 \pm 0.006$) (Fig. 6d, $p = 0.89$ and 0.36, respectively). The addition of ranolazine plus empagliflozin resulted in a FRET ratio of $1.42 \pm 0.006$, showing no significant difference from the addition of ranolazine alone ($1.42 \pm 0.006$) or empagliflozin alone (1.42) (Fig. 6d, $p = 0.91, 0.98$ each). These results, combined with the previous findings that empagliflozin directly inhibits NHE1 and Nav1.5, suggest that empagliflozin may enhance ATP levels in mitochondria via the inhibition of these ion channels or common downstream sites of these channels.

## Discussion

The current study was performed to evaluate the effect of empagliflozin on cardiac energy metabolism in T2D mice in order to directly quantify cytosolic and mitochondrial ATP levels. The hearts of empagliflozin-treated mice had significantly higher ATP levels in the cytoplasm and mitochondria after 10 weeks of treatment than control mice (*db/db*, 17 weeks old). Empagliflozin attenuated the decrease in and enhanced recovery of ATP levels at the ischemic site in an ischemia-reperfusion model of MI. Furthermore, the mitochondrial ATP level was increased 3 h after a single administration of empagliflozin. These results prompted us to examine cardiomyocytes derived from the model mice, and the

addition of 300 nM empagliflozin. The FRET ratio was $1.37 \pm 0.005$ in control and was significantly increased to $1.39 \pm 0.007$ in MA-5-treated cardiomyocytes, which facilitates mitochondrial ATP production via ATP synthase oligomerization (Fig. 6d, $p = 0.046$). In empagliflozin-treated cardiomyocytes, the FRET/GFP ratio was significantly increased to $1.42 \pm 0.006$ (Fig. 6d, $p = 0.0012$). To investigate the effect of cariporide and ranolazine, inhibitors of NHE1 and Nav1.5, respectively, we measured the ATP level in mitochondria of cardiomyocytes 1 h after the addition of cariporide. The FRET/GFP ratio was significantly increased to $1.42 \pm 0.006$ after the addition of cariporide and to $1.42 \pm 0.006$ after the addition of ranolazine (Fig. 6d, $p = 0.0001, 0.006$ each). We then examined whether or

direct effects of empagliflozin on the mitochondria of cardio-myocytes were demonstrated.

## Dynamics of ATP in the heart visualized by GO-ATeam.
Empagliflozin has been shown to increase ATP in the heart of *db/db* mice[7,8]. However, since the measurement was performed by extracting tissues after sacrificing animals, it was difficult to follow changes over time and monitor ATP production in intracellular organelles. In the present study, we used *GO-ATeam* mice, which can be used to visualize ATP dynamics in vivo and thereby evaluate the effect of empagliflozin on myocardial energy metabolism in diabetic mice. In contrast to findings with the conventional method, we found that empagliflozin increased ATP in mitochondria as early as 3 h after treatment. We also showed in a time-dependent fashion that empagliflozin attenuated the reduction in cytosolic ATP level and also enhanced the recovery of ATP at the infarct site during the ischemic-reperfusion period, suggesting the robustness of myocardial energy metabolism, as discussed below. In vitro studies demonstrated the differential effects of empagliflozin in cytosol or mitochondria on ATP production. Thus, *GO-ATeam* mice enabled us to visualize ATP levels in real time, in a living state, at organellar levels, making it a powerful tool for investigating energy metabolism under various perturbed conditions (e.g., drug administration, gene knockout, disease conditions) in a variety of organs/cells.

## Energetic remodeling in the diabetic heart by empagliflozin.
Encouraged by the finding that empagliflozin increased mito-chondrial ATP in *db/db* mice, which suggested the robustness of its effect on the cardiac energy state, we monitored the changes in cytosolic ATP concentrations in the ischemic region of the ischemia-reperfusion model after 10 weeks of empagliflozin administration. The changes in the cytoplasmic ATP levels during ischemia and reperfusion were clearly ameliorated in the empagliflozin-treated group, thus demonstrating that the energy production had become robust in vivo. In vitro studies using isolated cells, by contrast, revealed that 1-h treatment with empagliflozin significantly increased mitochondrial ATP but not cytoplasmic ATP. Furthermore, it also significantly increased mitochondrial ATP in wild-type-derived cardiomyocytes. Furthermore, we showed that empagliflozin alleviated the changes in mitochondrial ATP levels in cardiomyocytes during low-oxygen exposure, suggesting that the energy production had become robust in vitro. These in vivo/in vitro results suggest that empagliflozin not only quickly increases ATP production in mito-chondria, but also increases ATP production in the entire cell in the long term. Such qualitative changes in myocardial energy metabolism, which we call "remodeling", may underlie the cardiovascular benefits of empagliflozin seen in clinical trials and support the ongoing clinical trial evaluating the effect of empagliflozin on patients after acute MI (EMPACT-MI)[15].

## Mechanism underlying the effects of empagliflozin at the cellular and molecular levels.
Since SGLT2, the molecular target of empagliflozin, is expressed specifically in the proximal tubules of the kidney but not in the heart[16,17], the primary target tissue and target molecule for its cardioprotection has long been discussed. In the present study, we showed that empagliflozin rapidly reached the cardiac mitochondria in vivo. Furthermore, in vitro studies revealed that empagliflozin immediately increased mitochondrial ATP in isolated cardiomyocytes, suggesting that empagliflozin has a direct effect on the myocardium. In addition, our findings also showed that the cytosolic ATP levels did not increase in vitro, while the mitochondrial ATP levels and membrane potential both increased. These in vivo/in vitro results

suggest that the target in cardiomyocytes may reside in the mitochondria.

Some researchers have proposed the possibility of NHE1 and Nav1.5 as off-targets of empagliflozin in the heart[13,14]. In our study, we found that both cariporide and ranolazine, inhibitors of NHE1 and Nav1.5, respectively, increased mitochondrial ATP production. Empagliflozin, however, did not induce any additional benefit, suggesting that empagliflozin may increase mitochondrial ATP by inhibiting both NHE1 and Nav1.5. This conclusion is consistent with reports that NHE1 is expressed in mitochondrial membranes, with its inhibition maintaining the membrane potential[18], and that Nav1.5 is functionally linked to mitochondria in cardiomyocytes[19]. However, this should be interpreted with caution: first, a report was recently published arguing that empagliflozin does not inhibit NHE1[20]. If this is the case, to reconcile these findings with our own, the mechanism underlying the effects of empagliflozin may involve the common pathway of both inhibitors, i.e., reduction in $Ca^{2+}$ overload by decreasing the intracellular $Na^+$ concentration. Second, cariporide and ranolazine used as tool compounds in this study may not have been specific to each target[21–23], and the concentrations used are in the micromolar range; therefore, we cannot exclude the possibility that ATP is induced by their off-target effects. Further studies are needed to identify bona-fide molecular targets of empagliflozin in the heart.

## Strengths and limitations.
Several strengths and limitations associated with the present study warrant mention. First, our ATeam mouse is a very useful tool for visualizing energy metabolism in vivo, even at organellar levels, which is not limited to the heart. It should be pointed out, however, that this assay only measured cardiac surface cells, so the results need to be validated using two-photon microscopy or fiber microscopy, which can measure deeper areas. Second, we clearly showed the energetic effects of empagliflozin in the failing heart of diabetic *db/db* mice, which recapitulate the features of cardiomyopathy and heart failure[24,25]. However, given that there are many etiologies of heart failure and the clinical evidence indicating that empagliflozin is effective against heart failure with or without diabetes[26,27], to what extent our findings can be applied remains to be seen. Third, our in vitro experiments were conducted with cardiomyocytes isolated from the heart, so the molecular mechanism underlying ATP enhancement by empagliflozin that we postulated here is not necessarily the same as that seen in the in vivo native environment.

## Methods

### Chronic treatment of *db/db*; cytoATP-Tg mice or *db/db*; mitoATP-Tg mice with empagliflozin.
Seven-week-old *db/db*; cytoATP-Tg/mitoATP-Tg or control (+/+; cytoATP-Tg/mitoATP-Tg) mice were fed a diet with/without 30 mg/kg empagliflozin for 10 weeks. The dose of empagliflozin was chosen based on the data from previously published pre-clinical studies[7,10].

Prior to starting the experiments, baseline urine and blood samples were collected. The body weight and systolic blood pressure in awake mice were measured. The body weight was measured 8 weeks after the start of drug treatment, and the systolic blood pressure was measured 8 weeks after the start of drug treatment. Blood samples were collected from the tail vein of mice for measurement of non-fasting blood glucose and ketone levels at eight weeks after the start of drug treatment, and the blood pressure was measured at 8 weeks after the start of drug treatment. After 8 weeks of drug treatment, mice were housed in metabolic cages to collect 24-h urine. At the end of the 10-week treatment, 17-week-old mice were anaesthetized with isoflurane, and the chest was opened surgically. The beating heart was observed under a fluorescent microscope for ATP imaging in vivo.

Regarding the observation method, in brief, the object was exposed to an excitation light (ET470/40) using a fluorescence microscope (Leica M165FC; Leica Microsystems, Germany), and the absorbed light was separated into GFP (D515/30m, Chroma Technology, USA) and RFP (D575/40m, Chroma Technology, USA) by a Dual-View (DM540; Nippon Roper, Japan), and the image was captured by a

cMOS camera (ORCA-Flash4.0; Hamamatsu Photonics, Japan) to obtain dual-images simultaneously.

Analyses were performed using the MetaMorph software program (Molecular Devices, USA) and displayed as IMD images. For cytosolic ATP, since we were able to make a calibration curve showing the relationship between the FRET/GFP ratio and ATP level, we were able to estimate the ATP level from the FRET/GFP ratio (paper submission). However, for mitochondrial ATP, only the FRET/GFP ratio was described, as it is technically difficult to make a calibration curve.

All male mice used were 8 weeks old at the start of each experiment and were housed individually or in groups of 2–3 per cage at a temperature of 22 ± 1 °C with a 12-h light–dark cycle and ad libitum access to food and water. All procedures were performed in accordance with the guidelines of the Laboratory Animals Care and Use Committee (No. 20073, 21053, 22029). Efforts were made to minimize the number of animals used and to limit their suffering.

**Operation of ischemia and reperfusion in the heart**. After anesthesia with iso-flurane, overnight-fasted 17-week-old *db/db*; cytoATP-Tg mice were intubated and artificially ventilated. After opening the chest using electrocautery, ischemia was induced by ligating the left anterior descending artery with a suture needle and surgical thread (L6-80 n3; Natsume Seisakusho, Japan) under a microscope while monitoring the electrocardiogram (ECG). Reperfusion was performed by removing the suture 30 min after the creation of the infarction while monitoring the ECG. In vivo ATP imaging was performed using a fluorescence microscope (Leica M165FC) at six time points: before ligation (0 min), after ligation (20 and 30 min), just after reperfusion, and 15 and 30 min (45 and 60 min, respectively) after reperfusion.

**Transient treatment of *db/db*; mitoATP-Tg mice with empagliflozin**. Eight-week-old *db/db*; mitoATP-Tg or control mice received a single dose of empagli-flozin (30 mg/kg body weight) via oral gavage after 4 h of fasting. After another 3 h of fasting, the mitochondrial ATP level of the heart was monitored in vivo under fluorescent microscopy.

**Isolation of mature cardiomyocytes**. We isolated mature cardiomyocytes from the indicated genotype mice using a Langendorff perfusion system (Radnoti, USA). To investigate the effect of empagliflozin, cariporide, and ranolazine on cardio-myocyte energetics in mitochondria or cytosol, we studied the mitochondrial or cytosolic ATP levels of cardiomyocytes with/without empagliflozin (10, 100, 300, 1000 nM), cariporide (10 μM), ranolazine (30 μM) or MA-5 (10 μM). To investi-gate the effect of empagliflozin on the mitochondrial membrane potential of car-diomyocyte, we used the JC-1 with empagliflozin (300 nM). Two-photon microscopy (Leica SP8MP) was used to observe fluorescence.

**The quantification of empagliflozin in mitochondria**. A single oral dose of empagliflozin was administered to 8-week-old wild-type mice, and 3 h later, they were sacrificed, with the heart, kidney, liver, small intestine and brain tissue col-lected. The tissues were homogenized using a homogenizer (Nippi Inc.) containing mitochondria buffer (250 mM sucrose, 20 mM HEPES-KOH, 5 mM $KH_2PO_4$, 50 μM $MgCl_2$, 0.2% BSA, pH 7.5) and lysed by being passed through a 1-mL syringe with a 27-G needle 10 times. Homogenized lysates were next centrifuged at $800 \times g$ for 5 min at 4 °C, and supernatants were further centrifuged at $6000 \times g$ for 15 min at 4 °C. The resulting pellets containing mitochondria were resuspended in a mitochondria buffer. Protein concentrations were determined using the BCA protein assay (Thermo Fisher Scientific) according to the manufacturer's instructions.

**Hypoxic treatment of cardiomyocytes**. Cardiomyocytes were isolated from 18-week-old db/db; mitoATP-Tg, treated with empagliflozin (1000 nM) or DMSO in Tyrode solution for 1 h, and then transferred to Tyrode saturated with 95% $O_2$, 5% $CO_2$ for imaging of the mitochondrial ATP level using two-photon microscopy (0 min). Subsequently, hypoxia treatment was performed by switching from 95% $O_2$, 5% $CO_2$ state to 10% $O_2$, 5% $CO_2$, 85% $N_2$ state, and the mitochondrial ATP level was imaged by two-photon microscopy after 20 and 30 min. Further imaging was performed 15 and 30 min after switching again to 95% $O_2$ and 5% $CO_2$ (45 and 60 min, respectively).

**Statistical and reproductivity**. Data are expressed as the mean ± SEM. Statistical analyses were performed using the unpaired two-tailed Student's *t*-test to compare two groups and an analysis of variance (ANOVA) (*$p < 0.05$; **$p < 0.01$; NS, not significant). All data were normally distributed, and variance was similar between groups. Biological replicates were derived using different samples derived from different mice. The results represent data from at least three independent experi-ments. We estimated the required sample size considering the variation and mean of the samples. We used the fewest animals required to draw statistically valid conclusions. Our protocol required excluding mice if we observed an abnormal size, weight, or both or disease symptoms before performing experiments. How-ever, this was unnecessary, as all mice were phenotypically normal and healthy.

**Reporting summary**. Further information on research design is available in the Nature Portfolio Reporting Summary linked to this article.

## Data availability
The datasets generated during and/or analyzed during the current study are available from the corresponding author on reasonable request.

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

## Acknowledgements
We would like to thank Daiki Kinoshita and Hiroo Washio for their technical support, and Dr Atsutaka Yasui for the valuable discussion of the results. We thank Japan Medical Communications for the English-language proofreading. This work was supported by Grant-in-Aids for Scientific Research and the Japan Agency for Medical Research and Development (24116703, JPMJPR14MF, 19H03561, AMED-CREST Grant Number JP21gm1210011) (all funding to M. Yamamoto), (AMED-CREST Grant Number 21gm1210009h0003 funding to M. Yanagita). M Yanagita and M Yamamoto received grants from BI.

## Author contributions
Y. Kotobuki., M. Yanagita and M. Yamamoto designed the experiments. M. Yamamoto supervised the project. J.C., N.M., D.S., R.Y., Y. Kitai., A.O. and M. Yamamoto performed experiments. J.C., N.M., R.Y., Y. Kitai. and M. Yamamoto analyzed the data. J.C., N.M., D.W., Y.O., Y. Kotobuki, Y.N., M.P.P., H.I. and M. Yamamoto discussed the results. N.M. and M. Yamamoto wrote the manuscript.

## Competing interests
The authors declare the following competing interests: M. Yamamoto holds a patent for GO-ATeam mice. Y. Kotobuki, Y.N. and M.P.P. are employees of Boehringer Ingelheim. All other authors declare no competing interests.
