## [Peer Review File · Communications Biology]

Reviewers' comments:

Reviewer #1 (Remarks to the Author):

In the manuscript, Choi et al image the effect of SGLT2 inhibitor, empagliflozin, on cytosolic and mitochondrial ATP within db/db mouse cardiomyocytes, using a recombinant FRET sensor ATeam. In the Authors' hands, empagliflozin elevated both cytosolic and mitochondrial ATP in cardiomyocytes and overall imposed a cardioprotective effect. They argue that the effect could be mediated by decreased pressure on sodium handling, which sounds quite logical.

The work makes a very positive first impression: well-planned, well-written, figures perfectly arranged. The message is clear and the direct modality of the effect proven in isolated cardiomyocytes. I do 'buy' the main message, overall, although, I have to say, further reading brought multiple reservations, of which three look quite substantial to me.

1. Autofluorescence. There is a clear spectral overlap between the fluorescence of the chosen FRET ATP reporter and the (auto)fluorescence of the endogenous reduced equivalent, FAD (pls see e.g. here for the FAD spectrum: <https://www.becker-hickl.com/wp-content/uploads/2019/01/metabol-img-spectra-comp10-768x553.jpg>). With the excitation wl used (470 nm), FAD emission peak contributes to the GFP channel (515/30 nm) much more than to the RFP channel (575/40). The activation of the oxidative metabolism that leads to a decrease of FAD as it gets reduced to a non-fluorescent form, FADH, would correspond perfectly to the seeming increase of the RFP/GFP FRET ratio. Overall, the ratio will be dominated by the GFP signal, due to a 3-fold difference in the quantum yield vs RFP. To mend the problem, one could have performed some control experiments in the mouse heart that does not express the sensor and present the lack of the detectable autofluorescence signal or the alteration of that. I am mildly hopeful here, as the authors clearly used a low-NA objective for their imaging, which is less likely to pick up the autofluorescence. Nevertheless, the whole-organ autofluorescence can be quite substantial.

2. Static measurements of ATP using the FRET sensor (e.g. as in Figure 1: C vs B,D). I do not think one can use the FRET ratio to quantify the static/basal levels of ATP. The confounding factor here is the expression level of the sensor. The latter could be higher in the diabetic model as the substrate is highly available and apparently heart does not suffer much from insulin insensitivity. The level of expression can affect the perceived brightness of GFP and RFP in a different way and hence twist the seeming ATP readout. Complicating matters, the two fluorophores have very distinct quantum yields, as above. I believe one needs to perform real-time imaging experiments (as in figure 3), with basal level of sensor fluorescence, stimulated level (by glucose or methylated pyruvate) and inhibited level (oligomycin or FCCP) to draw any conclusions about true basal ATP level.

3. The pharmacology (figure 5) seemingly has no interaction with the empagliflozin effect. I am therefore not sure if it tells us anything about the mechanism. One could have used glucose analogues to alter the sodium flux via SGLT2, perhaps...

Technical points:

Abstract: needs rewriting, not fully academic style.

Figure 1: what does empa do to the control (panel E)?

Figure 2: how do we know the localisation is mitochondrial (panels F,G from figure 4 could be used here)? what does empa do to the control (panel E)?

Figure 3: impressive!

Figure 4: panel H could have looked more impressively as a dose-response curve

All figures: need scale bars.

Methods: microscopy/optics is not detailed well, as well as the timelapse imaging is not described. Optionally, a video of the timelapse (figure 3) is very much welcome.

Reviewer #2 (Remarks to the Author):

This study used mice that can directly measure cytosolic and mitochondrial ATP levels using modified ATeam, crossed with db/db mice to examine what effects empagliflozin, a sodium-glucose co-transporter 2 inhibitor, has on cardiac energetics. Empagliflozin treatment significantly increased cytosolic and mitochondrial ATP levels in the hearts of db/db mice. It also prevented the decrease in cytosolic ATP levels in the infarcted region of heart. Furthermore, empagliflozin restored the cytosolic ATP decrease in T2D heart by increasing mitochondrial ATP through NHE1 and Nav1.5. It is concluded that remodeling of energy metabolism and increased cardiac robustness may be involved in the mechanism of empagliflozin on the beneficial effects on heart failure.

General Comments:

This study uses novel cytoATP-Tg or mitoATP-Tg mice to examine cytoplasmic and mitochondrial levels in vivo. However, the authors provide no details on these mice, as it is the subject of another submitted paper. In addition, in this study the authors crossed cytoATP-Tg or mitoATP-Tg mice with db/db mice to assess the effects of empagliflozin on the cardiac energy status of those mice. However, more information on phenotype of the mice is needed, including whether cardiac function was normal or impaired between groups. There is also no data on whether empagliflozin is having any positive effect on cardiac function in these mice.

The authors highlight that to “investigate the mechanism underlying the improvement in cardiac energy status with empagliflozin, real-time and accurate monitoring of the pathways and amount of energy production in vivo is needed.” While this is true, the methodology developed is not really looking at the pathways, or the amount of energy production. It is just looking at cytoplasmic and mitochondrial high energy phosphate levels.

The conclusions that EMPA is working via the Na/H exchanger or Nav1.5 is weak (as described below).

I have a number of significant problems with the experimental data, which is highlighted below:

- 1) Very little information on the cytoATP-Tg/mitoATP-Tg mice. The manuscript describing this has not been published.
- 2) The mechanism by which EMPA improves ATP is missing from the acute fasting study.
- 3) The authors are selective in presenting cyto and mito ATP data. Mito ATP levels are missing from the infarct studies, and cyto are missing from the fasting mice.
- 4) If EMPA is acting through Na/H exchange or Nav1.5, how would this preserve ATP during ischemia?
- 5) Figure 1: Why were mice subjected to overnight fasting before ATP measurements? This could change energetics.
- 6) Figures should show individual data points.
- 7) Figure 1E and 2E: The y axis is misleading and is presented in such a way as to exaggerate differences (ie. by starting at 1).
- 8) Figure 1: The representative hearts shown in Figure 1B and D look very different. However, the FRET/GTP quantified looks the same. Are the figures really representative?
- 9) Figure 3: The data suggests that at the end of the reversible infarct (before reperfusion) that EMPA markedly preserved ATP levels in the infarct zone. I am not sure how this could be, if the two groups were equally ischemic. Was the ATP originating from glycolysis? Any lactate measurements?
- 10) Figure 3: The FRET/GFP image in the non-infarcted region of the db/db EMPA mice (Figure 3F) looks much stronger than the non-infarcted region of the db/db control mice. I am not sure why this would be? Figure 3C in the non-infarcted region looks much less intense than the signal in Figure

1C.

11) What happens to mito ATP levels in the infarcted hearts? No data is presented.

12) Figure 4E and 4H: The y axis is misleading and is presented in such a way as to exaggerate differences (ie. by starting at 1).

13) Figure 4C and 4D: What is the difference between these two hearts? Why are you separating out hearts based on "with or without increased ATP"?

14) Figure 4: Sample sizes are very low (n =3). Why?

15) The authors use isolated mature cardiomyocytes from 8-week-old db/db; mitoATP-Tg mice and examined the mitochondrial ATP levels in the same cells before and 1 h after the addition of empagliflozin. It is not clear, however, how could EMPA directly increase ATP levels?

16) Figure 5C and 5D: The y axis is misleading and exaggerates differences.

17) Figure 5: The authors do not adequately address the controversy that EMPA does not modify Na/H exchange activity in the heart.

18) The lack of a synergistic effect of EMPA and cariporide does not necessarily imply that EMPA is acting via the Na/H exchanger.

19) The lack of a synergistic effect of EMPA and ranolazine does not necessarily imply that EMPA is acting via Nav 1.5. At the dose of ranolazine used, there is also effects of ranolazine on energy metabolism.

First of all, we want to thank both the reviewers for their critical reading of our manuscript and their constructive comments. In the following, we answered the reviewer's comments one by one.

Referee: 1

1) Autofluorescence. There is a clear spectral overlap between the fluorescence of the chosen FRET ATP reporter and the (auto)fluorescence of the endogenous reduced equivalent, FAD (pls see e.g. here for the FAD spectrum: <https://www.becker-hickl.com/wp-content/uploads/2019/01/metabol-img-spectra-comp10-768x553.jpg>). With the excitation wl used (470 nm), FAD emission peak contributes to the GFP channel (515/30 nm) much more than to the RFP channel (575/40). The activation of the oxidative metabolism that leads to a decrease of FAD as it gets reduced to a non-fluorescent form, FADH, would correspond perfectly to the seeming increase of the RFP/GFP FRET ratio. Overall, the ratio will be dominated by the GFP signal, due to a 3-fold difference in the quantum yield vs RFP. To mend the problem, one could have performed some control experiments in the mouse heart that does not express the sensor and present the lack of the detectable autofluorescence signal or the alteration of that. I am mildly hopeful here, as the authors clearly used a low-NA objective for their imaging, which is less likely to pick up the autofluorescence. Nevertheless, the whole-organ autofluorescence can be quite substantial.

We have modified the manuscript and the figure taking the reviewer's comment into consideration, i.e. we mentioned that the effect of autofluorescence in the heart was small enough to be negligible.

[Please see p.7-8, line 98-105 and Supplementary Fig. 2a-m]

2) Static measurements of ATP using the FRET sensor (e.g. as in Figure 1: C vs B,D). I do not think one can use the FRET ratio to quantify the static/basal levels of ATP. The confounding factor here is the expression level of the sensor. The latter could be higher in the diabetic model as the substrate is highly available and apparently heart does not suffer much from insulin insensitivity. The level of expression can affect the perceived brightness of GFP and RFP in a different way and hence twist the seeming ATP readout. Complicating matters, the two fluorophores have very distinct quantum yields, as above. I believe one needs to perform real-time imaging experiments (as in figure 3), with basal level of sensor fluorescence, stimulated level (by glucose or methylated

pyruvate) and inhibited level (oligomycin or FCCP) to draw any conclusions about true basal ATP level.

We have modified the manuscript and the figure taking the reviewer's comment into consideration, i.e. we referred not to the ATP level but to the FRET/GFP ratio as the indicator of the amount of ATP.

[Please see p.8, line 106-107 and Fig. 1a-e]

3) The pharmacology (figure 5) seemingly has no interaction with the empagliflozin effect. I am therefore not sure if it tells us anything about the mechanism. One could have used glucose analogues to alter the sodium flux via SGLT2, perhaps...

We agree that our results just showed empagliflozin did not induce any additional effect on the effect of cariporide or ranolazine alone, not necessarily meaning that the mechanism of the empagliflozin effect is the same as that of cariporide or ranolazine. Hence, we have added discussion regarding this point.

[Please see p. 19 line 306-321]

Technical points:

1) Abstract: needs rewriting, not fully academic style

We have followed the reviewer's suggestion and have modified the Abstract part.

[Please see p.3, line 30-42].

2) Figure 1: what does empa do to the control (panel E)?

We have confirmed that empagliflozin treatment did not affect the ATP level in the heart in wild-type mice.

3) Figure 2: how do we know the localisation is mitochondrial (panels F,G from figure 4 could be used here)? what does empa do to the control (panel E)?

We have modified the manuscript and referred the biodistribution of empagliflozin in mitochondria in various tissues, suggesting that empagliflozin accumulated in mitochondria in the heart and other organs, except for the brain.

[Please see p.11, line 173-174 and Fig. 4h]

4) Figure 4: panel H could have looked more impressively as a dose-response curve

We have added the data of the group with 500 nM empagliflozin administration and have clarified its dose-dependent effect.

[Please see p.12, line 183-187 and Fig. 4g]

5) *All figures: need scale bars.*

We have added scale bars to all figures.

6) *Methods: microscopy/optics is not detailed well, as well as the timelapse imaging is not described. Optionally, a video of the timelapse (figure 3) is very much welcome.*

We have added the details about microscopy/optics and timelapse imaging in the Methods part. The ATP levels of the heart were measured every thirty minutes in the experiment (Fig. 2). Therefore, we have to say that we cannot offer a video of the timelapse.

[Please see line 354-359 (p.21-22), 378-381 (p.23), 395-396 (p.24), 410-417 (p.24-25)]

Referee: 2

1) *Very little information on the cytoATP-Tg/mitoATP-Tg mice. The manuscript describing this has not been published.*

The cytoATP-Tg mice are detailed in our previous study mentioned below. We have added the explanation of experiments using these mice in the Methods part.

[Please see p.22, line 361-365 and “Koitabashi, Norimichi, Riki Ogasawara, Ryuto Yasui, Yuki Sugiura, Hinako Matsuda, Shigenori Nonaka, Takashi Izumi, et al. 2020. “Visualizing ATP Dynamics in Live Mice.” *bioRxiv*.

<https://doi.org/10.1101/2020.06.10.143560>.”]

2) *The mechanism by which EMPA improves ATP is missing from the acute fasting study.*

We would like first to clarify that when it comes to the cardioprotective mechanism of EMPA, our results just showed empagliflozin rapidly accumulated in the cardiac mitochondria (Fig. 4h) and did not induce any additional effect on the effect of cariporide or ranolazine alone (Fig. 6). Accordingly, we acknowledge that we cannot conclude the detailed mechanism of its cardioprotective effect, and hence we have added discussion regarding this point.

[Please see p.19, line 313-321]

3) *The authors are selective in presenting cyto and mito ATP data. Mito ATP levels are missing from the infarct studies, and cyto are missing from the fasting mice.*

We have taken the reviewer's suggestion into consideration and performed the serial measurement of mitochondrial ATP levels in mature cardiomyocytes isolated from *db/db*; mitoATP-Tg mice during hypoxia exposure and oxygen recovery (Fig. 5). We have also added information about the effect of single treatment with empagliflozin on cytosolic ATP levels in the heart (Sup Fig. 4). We have to say that it is technically challenging to perform the serial measurement of ATP levels at infarct sites of the heart of *db/db*; mitoATP-Tg mice for the following reasons. The *db/db*; mitoATP-Tg mice used in the experiment can be generated only with the probability of 1/32 by crossing mitoATP-Tg mice with a mouse model of T2D. In addition, more than half of the mice are not capable of bearing the load of operation of ischemia and reperfusion.

[Please see line 187-188, 195-211(p.12-13), Fig. 5, and Sup Fig. 4]

4) *If EMPA is acting through Na/H exchange or Nav1.5, how would this preserve ATP during ischemia?*

As mentioned above, we agree that our results just showed empagliflozin did not induce any additional effect on the effect of cariporide or ranolazine alone (Fig. 6), not necessarily meaning that the mechanism of the empagliflozin effect is the same as that of cariporide or ranolazine. Accordingly, we have to say that we cannot explain more detailed mechanism of the preservation of ATP levels provoked by EMPA during ischemia, and hence we have added discussion regarding this point.

[Please see p.19, line 313-321]

5) *Figure 1: Why were mice subjected to overnight fasting before ATP measurements? This could change energetics.*

The purpose of overnight fasting is to prevent when each mouse eats the diet from affecting the ATP levels in their hearts.

6) *Figures should show individual data points.*

All raw data used to make graphs are available in the main text or the supplementary materials. Hence, we thought that individual data points were not necessarily required in all figures.

7) Figure 1E and 2E: The y axis is misleading and is presented in such a way as to exaggerate differences (ie. by starting at 1).

Regarding the bar charts, the y axes were tailored to the bars indicating the range of FRET/GFP ratio in the fluorescence images.

8) Figure 1: The representative hearts shown in Figure 1B and D look very different. However, the FRET/GFP quantified looks the same. Are the figures really representative?

We are sorry for the discrepancy between the images (Fig. 1B, D) and the graph (Fig. 1E). We have corrected the images accordingly to avoid misunderstandings. [Please see Fig. 1b, d]

9) Figure 3: The data suggests that at the end of the reversible infarct (before reperfusion) that EMPA markedly preserved ATP levels in the infarct zone. I am not sure how this could be, if the two groups were equally ischemic. Was the ATP originating from glycolysis? Any lactate measurements?

As mentioned above, when it comes to the cardioprotective mechanism of EMPA, our results just showed empagliflozin rapidly accumulated in the cardiac mitochondria (Fig. 4h) and did not induce any additional effect on the effect of cariporide or ranolazine alone (Fig. 6). Accordingly, we have to say that we cannot explain more detailed mechanism of the preservation of ATP levels provoked by EMPA in the infarct region, and hence we have added discussion regarding this point.

[Please see p.19, line 313-321]

10) Figure 3: The FRET/GFP image in the non-infarcted region of the db/db EMPA mice (Figure 3F) looks much stronger than the non-infarcted region of the db/db control mice. I am not sure why this would be? Figure 3C in the non-infarcted region looks much less intense than the signal in Figure 1C.

The following points would be given as reasons. After hypoxia was first observed in the LAD region, it steadily spread to the peripheral region. As a result, the non-infarcted region of the heart shown in Figure 2c looks much less intense than the signal in Figure 1c. In addition, the FRET/GFP signal in the non-infarcted region of the db/db EMPA mice (Figure 2f) looks much stronger than the non-infarcted region of the db/db control mice because the decrease in ATP levels was

suppressed in the whole heart including the peripheral region in the EMPA group compared to the control group due to the chronic treatment with EMPA.

11) What happens to mito ATP levels in the infarcted hearts? No data is presented.

As mentioned above, we have performed the serial measurement of mitochondrial ATP levels in mature cardiomyocytes isolated from *db/db*; mitoATP-Tg mice during hypoxia exposure and oxygen recovery (Fig. 5).

We have to say that it is technically challenging to perform the serial measurement of ATP levels at infarct sites of the heart of *db/db*; mitoATP-Tg mice for the following reasons. The *db/db*; mitoATP-Tg mice used in the experiment can be generated only with the probability of 1/32 by crossing mitoATP-Tg mice with a mouse model of T2D. In addition, more than half of the mice are not capable of bearing the load of operation of ischemia and reperfusion while imaging.

[Please see p.12-13, line 195-211 and Fig. 5]

12) Figure 4E and 4H: The y axis is misleading and is presented in such a way as to exaggerate differences (ie. by starting at 1).

Regarding the bar charts, the y axes were tailored to the bars indicating the range of FRET/GFP ratio in the fluorescence images.

13) Figure 4C and 4D: What is the difference between these two hearts? Why are you separating out hearts based on “with or without increased ATP”?

We are sorry for the confusing images and have modified the figure and its legend to clarify the difference between the EMPA and control groups and avoid misunderstandings.

[Please see p.36 line 565-567 and Fig. 4b, c]

14) Figure 4: Sample sizes are very low (n =3). Why?

We have taken the reviewer’s comment into consideration and implemented the additional experiments to make the sample size sufficient, modifying the figure legend accordingly.

[Please see p.36 line 567-568]

*15) The authors use isolated mature cardiomyocytes from 8-week-old *db/db*; mitoATP-Tg mice and examined the mitochondrial ATP levels in the same cells before and 1 h*

after the addition of empagliflozin. It is not clear, however, how could EMPA directly increase ATP levels?

As mentioned above, when it comes to the cardioprotective mechanism of EMPA, our results just showed empagliflozin rapidly accumulated in the cardiac mitochondria (Fig. 4h) and did not induce any additional effect on the effect of cariporide or ranolazine alone (Fig. 6). Accordingly, we have to say that we cannot explain more detailed mechanism of the increase of mitochondrial ATP levels directly caused by EMPA in the isolated mature cardiomyocytes, and hence we have added discussion regarding this point.

[Please see p.19, line 313-321]

16) Figure 5C and 5D: The y axis is misleading and exaggerates differences.

Regarding the bar charts, the y axes were tailored to the bars indicating the range of FRET/GFP ratio in the fluorescence images.

17) Figure 5: The authors do not adequately address the controversy that EMPA does not modify Na/H exchange activity in the heart.

We acknowledge the possibility that empagliflozin does not modify Na/H exchange activity in the heart. We have added the discussion regarding this point accordingly.

[Please see p.19, line 313-321]

18) The lack of a synergistic effect of EMPA and cariporide does not necessarily imply that EMPA is acting via the Na/H exchanger.

and

19) The lack of a synergistic effect of EMPA and ranolazine does not necessarily imply that EMPA is acting via Nav 1.5. At the dose of ranolazine used, there is also effects of ranolazine on energy metabolism.

As mentioned above, we agree that our results just showed empagliflozin did not induce any additional effect on the effect of cariporide or ranolazine alone, not necessarily meaning that the mechanism of the empagliflozin effect is the same as that of cariporide or ranolazine. Hence, we have added discussion regarding this point.

[Please see p.19, line 313-321]

REVIEWERS' COMMENTS:

Reviewer #1 (Remarks to the Author):

The Authors have adequately dealt with my suggestions/concerns.

Reviewer #2 (Remarks to the Author):

none